# MINT: MATRIX-INTERLEAVING FOR MULTI-TASK LEARNING

## ABSTRACT

Deep learning enables training of large and flexible function approximators from scratch at the cost of large amounts of data. Applications of neural networks often consider learning in the context of a single task. However, in many scenarios what we hope to learn is not just a single task, but a model that can be used to solve multiple different tasks. Such multi-task learning settings have the potential to improve data efficiency and generalization by sharing data and representations across tasks. However, in some challenging multi-task learning settings, particularly in reinforcement learning, it is difficult to learn a single model that can solve all the tasks while realizing data efficiency and performance benefits. Learning each of the tasks independently from scratch can actually perform better in such settings, but it does not benefit from the representation sharing that multi-task learning can potentially provide. In this work, we develop an approach that endows a single model with the ability to represent both extremes: joint training and independent training. To this end, we introduce matrix-interleaving (Mint), a modification to standard neural network models that projects the activations for each task into a different learned subspace, represented by a per-task and per-layer matrix. By learning these matrices jointly with the other model parameters, the optimizer itself can decide how much to share representations between tasks. On three challenging multi-task reinforcement learning problems with varying degrees of shared task structure, we find that this model consistently matches or outperforms joint training and independent training, combining the best elements of both.

## 1 INTRODUCTION

While deep learning has enabled remarkable levels of generalization through the use of function approximators, this comes at the cost of large amounts of data, which remains a critical challenge in deploying deep learning to a number of domains. When combined with deep networks, multi-task learning offers the promise of building more powerful representations using less data per task, leading to greater performance and data efficiency. However, multi-task deep learning has also posed considerable challenges. Numerous works have observed that joint training on multiple tasks can actually *decrease* task performance due to the negative influence of other tasks (Parisotto et al., 2015; Rusu et al., 2016a). Indeed, training networks entirely independently on each task has remained a strong approach, to the point that multiple multi-task methods have first trained models independently before using them to train a multi-tasking model (Parisotto et al., 2015; Rusu et al., 2016a; Ghosh et al., 2017; Teh et al., 2017; Czarnecki et al., 2019). Moreover, our experiments in Section 6 indicate that three recently proposed methods for multi-task learning are all surpassed by training models independently per task. However, training independent models will only work well when provided enough data per task, and precludes potential positive data-efficiency gains from multi-task learning, only providing protection against negative transfer. Further, while a number of works have successfully shared parameters, finding an architecture with the appropriate level of parameter sharing for a given problem domain can require a considerable amount of manual engineering. In this work, we aim to develop a multi-task learning method that can perform well both when tasks share very little and when they share a large amount of structure.

To address this problem, we consider how a single neural network model can represent two extremes: independent models, when optimization challenges prevail, or a single model with shared weights, when sharing is beneficial. Further, we would like such a model to be able to represent intermediate

levels of model sharing, when appliable. One option for performing independent training within a single model is to put separate networks with independent weights into a single model, using the task ID to select which network prediction to output. However, this prevents any sharing. An alternative approach is to condition the model on the task ID, through various conditioning approaches, including additive and multiplicative approaches such as FiLM (Perez et al., 2018). In fact, point-wise multiplicative conditioning, as proposed in FiLM, can indeed represent separate networks by selecting which parts of the network to be used for different tasks, as can a number of other approaches in multi-task learning (Rosenbaum et al., 2017; 2019; Fernando et al., 2017). Yet, these approaches still require an optimization over shared parameters in order to select which parameters are used for each task. These shared parameters can introduce significant optimization challenges.

We instead consider how to allow a model to perform optimization on only shared parameters, only disjoint parameters, or any combination thereof. We can achieve this by simply interleaving learned per-task matrices at each layer of a jointly-trained neural network. When optimization over shared parameters is ineffective, the model can still represent a full neural network per task using only the per-task matrices, resulting in independent training; while using identical per-task matrices results in standard joint training. Intermediately, a mix of shared and per-task parameters may be used. In effect, by incorporating these matrices into the network, the optimizer itself can automatically and dynamically modulate the degree to which a representation is shared between tasks, depending on the problem domain and the optimization progress, and can do so *without* having to optimize shared parameters.

The primary contribution of this paper is a simple yet effective approach for multi-task learning that can represent and smoothly interpolate between independent training and joint training, via matrix interleaving (Mint). We describe how we can implement Mint in deep multi-task models and show its effectiveness in improving data efficiency and generalization in multi-task settings while providing intuition about the reasons why this architecture performs so well. Further, we show that the model can be extended to goal-conditioned reinforcement learning in a straightforward manner by allowing the model to generate the interleaved matrices conditioned on task information such as the goal. We evaluate Mint on sets of tasks with both high and low levels of shared structure and find that it performs well in both settings, performing comparably to or outperforming both joint training and independent training, effectively combining the best elements of both. Further, in comparison to previous methods that use multiplicative interactions for continual learning (Cheung et al., 2019) and for general conditioning (Perez et al., 2018), Mint is better able to separate tasks by avoiding the need to optimize over shared parameters and can *empirically* produce substantially better performance on a range of challenging multi-task problems. Finally, Mint also outperforms state-of-the-art approaches for multi-task learning while being significantly simpler to implement.

## 2 PRELIMINARIES

In multi-task learning, the goal is to find a $\theta$-parameterized model $f_\theta$ that reaches high performance across all training tasks drawn from a task distribution $\mathcal{T}_k \sim p(\mathcal{T})$, i.e. $\min_\theta \mathbb{E}_{\mathcal{T}_k \sim p(\mathcal{T})} [\mathcal{L}_k(f_\theta)]$, where $\mathcal{L}_k$ denotes the loss function for task $\mathcal{T}_k$. In this work, we study the setting in which we have a fixed set of $K$ tasks $\{\mathcal{T}_i\}_{i=1}^K$, and we wish to obtain high performance on all of the tasks in this set. In Section 4, we will study this multi-task problem in the context of a reinforcement learning setting. In our multi-task learning setup, we train a model that is conditioned on a task indicator $z_k$ which is used to specify a task $T_k$ that should be performed. The task indicator can be represented in a variety of ways, from simple categorical variables to learned task embeddings (Hausman et al., 2018). This formulation can be readily extended to a goal-conditioned reinforcement learning setting, where $z_k$ indicates the desired goal. The multi-task learning model learns to optimize objectives of all the tasks $T_k$ that the model is trained on.

Joint and fully independent training are the two extremes of multi-task learning. Assuming a set of $n$ training tasks, we characterize multi-task training as independent if it optimizes a set of task-specific, disjoint parameters $\{\theta^k\}_{i=1}^n$ that parameterize the model $f_{\theta^k}$. We define joint training as finding a single set of task-independent, shared parameters $\theta$ of $f_\theta$. Note that joint training utilizes the fact that the parameters are shared *throughout* learning. While joint training has a number of data-efficiency and generalization benefits, it can be difficult to train effectively.

## 3 MULTI-TASKING WITH INTERLEAVED MATRICES

Considering fully independent and fully joint training as two ends of the spectrum in multi-task learning problems, we want to design an algorithm to get the best of both worlds – the stability of independent training and the parameter-sharing efficiency of jointly trained models. To this end, we propose to build models that allow the neural network to learn how much information should be shared and between which tasks throughout learning, in an adaptive fashion. We describe the details of our approach below.

### 3.1 MINT: MATRIX-INTERLEAVED NETWORKS

In the case of neural network models, we view representations as network activations at different layers of the network. We aim to introduce a modification to the neural network that would allow the model to either form those activations in a completely task-specific way, in a completely shared way, or in a way that shares to an intermediate degree. To achieve this, we propose a model architecture that transforms the previous layer's representation both in a task-general way and in a task-specific way, in sequence. When two tasks share very little, the network can optimize task-specific weights, while when the tasks share a considerable degree of structure, the network can leverage the shared weights. Since these transformations are task-specific and can be introduced at various layers of the network, they allow for a different amounts of representation shared at different levels of the neural network model.

To understand the practical implementation, we consider the activations at layer $l$ of a fully-connected neural network: $y^{(l)} = \sigma\left(W^{(l)}y^{(l-1)} + b^{(l)}\right)$, where $W^{(l)}$ is the weight matrix for layer $l$, $b^{(l)}$ is the bias vector for layer $l$, and $\sigma$ is the non-linearity at each layer. The Mint layer augments the traditional fully-connected layer with task-specific weight matrix $M_k$ and bias vector $\beta_k$, where $k$ indexes the task. The forward pass of a Mint layer for some vector of activations $y$ is presented in Definition 1.

**Definition 1.** *A Mint layer applies an affine transformation to activations $y \in \mathbb{R}^n$ as follows, yielding new activations a:*

$$a = \texttt{Mint}(y) = M_k y + \beta_k \tag{1}$$

where $M_k$ and $\beta_k$ are per-layer *task-specific* matrix and bias, respectively. A neural network augmented with Mint thus contains parameters that are both shared across all tasks as well as parameters that are only used for a particular task for each layer $l$, i.e. $\theta = \{W^{(l)}, b^{(l)}, M_k^{(l)}, \beta_k^{(l)}\}$. See Figure 1 for a visual depiction of the application of Mint. We show how the regular fully-connected layers and Mint layers can be interleaved in Equation 2 and 3 below:

$$y^{(l)} = \sigma\left(W^{(l)}a^{(l-1)} + b^{(l)}\right) \tag{2}$$

$$a^{(l)} = M_k^{(l)}y^{(l)} + {\beta_k}^{(l)}. \tag{3}$$

Because this layer is fully-differentiable, we can learn the task-specific matrices $M_k^{(l)}$ and biases $\beta_k^{(1)}$ jointly with the shared parameters of the model.

When we apply Mint to tasks with very large numbers of tasks, or arbitrary task descriptors (e.g., goal-conditioning), we can train separate neural networks $T_M^l, T_\beta^l \to M_k^l, \beta_k^l$, to output the Mint matrices and biases at every layer, instead of storing independent matrices for each task. In this case, Mint resembles FiLM Perez et al. (2018), another method which performs transformation-based task conditioning. In contrast to FiLM, Mint uses a matrix transforming the activations at each layer instead of a point-wise multiplication by a vector. In the next section, we study a theoretical property of Mint that motivates the chosen definition of the Mint layer. We validate the benefits of Mint in our experimental evaluation in Section 6.

### 3.2 THE EXPRESSIVE POWER OF MINT

We next aim to theoretically study the expressive power of multi-task learning architectures when the shared parameters cannot be optimized effectively (e.g. due to optimization challenges arising from learning tasks that share little or no structure). While a viable approach in this regime is to simply use separate task-specific networks $f_{\phi_i}$ with no shared parameters, this approach makes the strong assumption that the tasks indeed share no exploitable structure. This is often not known a priori.

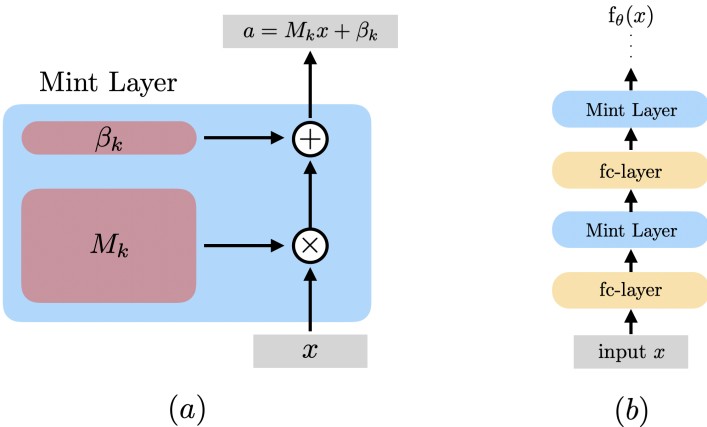

Figure 1: Mint architecture. (a) A Mint Layer applies a task-specific matrix multiplication and vector addition to an input vector $x$. The matrix and vector make up an affine transformation to the input vector, given by a task network. (b) The modified neural network model with a distinct Mint layer added after each fully-connected layer.

We show in this section that in the 'worst case' where shared parameters are not useful, the class of functions defined by a Mint network is still as rich as that defined by a set of purely task-specific networks. Thus, Mint makes no sacrifices in terms of expressiveness with respect to the optimal attainable parameters $\phi_i^*$ and retains the possibility of exploiting shared structure if it does exist. Further, we show that the same is not true of existing methods that are related to Mint. Specifically, we compare the class of functions representable by Mint with

1. The class of functions defined by a FiLM network (Perez et al., 2018), where a layer's activations are 'tilted' through element-wise multiplication with a task-specific vector $\gamma_k$ and addition with a task-specific bias $\beta_k$.

2. The class of functions expressable by a task indicator concatenation approach, where a one-hot task indicator vector $z_k$ for task $\mathcal{T}_k$ is concatenated to each shared layer's input.

We begin by considering the optimal parameters for task-specific networks $f_{\phi_i^*}$, where $\phi_i^* = \arg\min_\phi \mathcal{L}_i(\phi)$, the parameters that minimize the loss function $\mathcal{L}_i$ for task $\mathcal{T}_i$. We assume that $f_{\phi_i^*}$ is an $L$-layer MLP. We also consider $L$-layer Mint networks $f_{\theta_i}$ (consisting of $L$ sequential pairs of Mint transform and shared parameter layer as in Equation 1) and $L$-layer FiLM networks (consisting of $L$ sequential pairs of FiLM transform and shared parameter layer). We state Definition 2 to complete our setup.

**Definition 2.** *Let $y^{(l-1)}$ be the activations of layer $l - 1$, and let $\alpha_i = \{W^{(l)}, b^{(l)}\}$ be the shared parameters (weight matrix and bias vector) at the $l$-th layer. We assume that $W^{(l)}$ is an arbitrary, fixed invertible matrix.*

With this setup, we state Lemmas 1 and 2, which highlight a key difference in the layer-wise expressiveness of Mint and methods such as FiLM and task indicator concatenation.

**Lemma 1.** *For a given $\alpha_i$, applying Mint to $y^{(l-1)}$ can express an arbitrary affine transformation at layer $l$ for each task.*

**Lemma 2.** *(a) For a given $\alpha_i$, there exist affine transformations that cannot be expressed by applying FiLM to $y^{(l-1)}$. (b) Similarly, for a given $\alpha_i$, there exist affine transformations that cannot be expressed by a network which concatenates the input $y^{(l-1)}$ of each layer $l$ with a task indicator $z_k$.*

The proof can be found in Appendix A. We assume only that the shared parameter matrices $W^{(l)}$ are invertible, such that they do not lose information about the input. This assumption critically does not require the optimizer to productively update the shared parameters.

We use Lemmas 1 and 2 to extend the above layer-wise expressiveness properties of Mint and FiLM to compare the function classes defined by $L$-layer Mint networks $f_{\theta_i}$ and FiLM networks $f_{\psi_i}$ in the

context of optimal $L$-layer MLP $f_{\phi_i^*}$. We state this comparison in Theorem 1. We use $\alpha$ to denote the subset of $\theta_i$ and $\psi_i$ that corresponds to the parameters of the network that are shared across tasks.

**Theorem 1.** *For all $\alpha$, there exists $\theta_i^*$ such that $f_{\theta_i^*}(x) = f_{\phi_i^*}(x)$ for all $x$. Further, there exists $\alpha$ and $\bar{x}$ such that $f_{\psi_i}(\bar{x}) \neq f_{\phi_i^*}(\bar{x})$ for all $\psi_i$.*

Because $f_{\phi_i^*}$ is an MLP (a composition of affine transformations and activation functions), the proof of the first statement follows from applying Lemma 1 to each layer of $f_{\phi_i^*}$ and $f_{\theta_i}$ (the task specific and Mint networks, respectively). Similarly, Lemma 2 implies that we can construct an $\alpha$ and $\mathcal{L}_i$ (and therefore $\phi_i^*$) such that we can find an $\bar{x}$ satisfying the second statement.

That is, for any set of shared parameters in a Mint network and any set of optimal 'target' MLP, there exist weights for Mint that are effectively equivalent to the target MLP. On the other hand, there exist settings of the shared parameters that make matching the outputs of the target MLP unattainable in the FiLM and task indicator concatenation regimes. Using Lemma 2, we reach an equivalent conclusion regarding the limited expressiveness of the task indicator concatenation approach.

## 4 FLAVORS OF MINT

The general idea of Mint is implemented as follows in the supervised and reinforcement learning settings.

**Multi-Task Supervised Learning** In the case of supervised learning models, we simply apply Mint layers after every fully connected layer of the model. Specifically, for a task identifier $z_k \in \mathbb{R}^K$ where $K$ is the number of tasks and for every layer $l$ with activations $a^{(l)} \in \mathbb{R}^n$, nonlinearity $\sigma$ and weight $W^{(l)}$ and bias $b^{(l)}$, we represent the transformation using two matrices $T_M^{(l)} \in \mathbb{R}^{n \times n \times K}$ and $T_\beta^{(l)} \in \mathbb{R}^{n \times K}$ that take in the task identifier and output the per-layer task-specific matrices $M^{(l)}$ and biases $\beta^l$ respectively. The transformation can be summarized as follows:

$$a^{(l)} = (T_M^{(l)} z_k)\sigma(W^{(l)} a^{(l-1)} + b^{(l)}) + T_\beta^{(l)} z_k = M_k^{(l)} \sigma(W^{(l)} a^{(l-1)} + b^{(l)}) + \beta_k^{(l)}. \quad (4)$$

**Multi-Task Reinforcement Learning** For multi-task reinforcement learning, we implement the architecture similarity to the supervised learning case but we combine this with actor-critic RL algorithms by introducing this architecture into both the critic $Q(s, a, z_k)$ and the actor $\pi(a|s, z_k)$.

**Goal-Conditioned Reinforcement Learning** For the case of goal conditioned RL, we introduce a slightly modified Mint architecture into both the actor and the critic conditioned on the task goal $g$. Specifically, for every layer $l$ with activations $a^{(l)}$, nonlinearity $\sigma$, weight $W^{(l)}$, and bias $b^{(l)}$, we represent two transformation function $T_\phi^{(l)}$ and $T_\psi^{(l)}$ by two 2-layer ReLU networks that take in the goal and produces a per-layer goal-specific matrix $M^l(g)$ and the bias $\beta^{(l)}(g)$ respectively. The transformation can be summarized as:

$$a^{(l)} = T_\phi^{(l)}(g)\sigma(W^{(l)} a^{(l-1)} + b^{(l)}) + T_\psi^{(l)}(g) = M(g)^{(l)} \sigma(W^{(l)} a^{(l-1)} + b^{(l)}) + \beta(g)^{(l)}. \quad (5)$$

## 5 RELATED WORK

Multi-task learning (Caruana, 1997; Bakker & Heskes, 2003; Ruder, 2017) focuses on the problem of finding a single model that can solve multiple different tasks. This formulation can be readily applied to a variety of learning domains, such as supervised learning (Zhang et al., 2014; Long & Wang, 2015; Yang & Hospedales, 2016b; Sener & Koltun, 2018; Zamir et al., 2018), and multi-task (Espeholt et al., 2018; Wilson et al., 2007; Hessel et al., 2019) and goal-conditioned reinforcement learning (Kaelbling, 1993; Andrychowicz et al., 2017; Pong et al., 2018). While multi-task learning offers the promise of efficient training of shared representations, naïvely training a single model on multiple tasks often does not result in these desired benefits, due to the optimization challenges introduced by the multi-task setting (Teh et al., 2017; Ghosh et al., 2017; Rusu et al., 2016a).

In order to eliminate potential negative interference between different tasks during multi-task learning using a single model, many approaches propose to learn each task separately, to later combine their solutions into a single multi-task model (Levine et al., 2016; Teh et al., 2017; Ghosh et al., 2017; Rusu et al., 2016a; Czarnecki et al., 2019; Parisotto et al., 2015). In contrast to these works, we present a method that is able to train a single model on multiple tasks and is able to interpolate between the extremes of joint and independent training.

More closely related to our approach, various architectural solutions have been proposed to increase the multi-task learning capability of the model. Example approaches include architectural changes that allow multiple modules or paths within the same network (Fernando et al., 2017; Devin et al., 2016; Misra et al., 2016; Rusu et al., 2016b; Vandenhende et al., 2019; Rosenbaum et al., 2017), transformation-based task conditioning (Perez et al., 2018), attention-based architectures (Liu et al., 2018; Maninis et al., 2019), multi-headed network solutions (Long & Wang, 2015; Riedmiller et al., 2018; Wulfmeier et al., 2019), and a variety of other approaches (Hashimoto et al., 2016; Ruder12 et al., 2017). We demonstrate an approach that allows for a middle ground between the conceptual extremes of fully independent training at one end and single-model joint training at the other. This added flexibility enables us to sidestep the negative effects of potential task interference while at the same time share parameters between the tasks when appropriate. Prior approaches (Yang & Hospedales, 2016a; Long et al., 2017) have investigated this ability by factorizing layers in the neural network across independent and shared components. In contrast, our method is simpler to implement, less computationally intensive, and empirically easier to optimize. In our experiments, we provide direct comparisons of our method to cross-stich (Misra et al., 2016), routing networks (Rosenbaum et al., 2019), the FiLM architecture (Perez et al., 2018), rotational superposition (Cheung et al., 2019), and multi-headed models.

## 6 EXPERIMENTS

The goal of our experimental evaluation is to answer the following questions: (1) does our method enable effective multi-task learning both in settings where there is substantial overlap in tasks and where there is little overlap?, (2) how does our method compare to independent training and joint training in those settings?, (3) how does our method compare to prior state-of-the-art approaches?

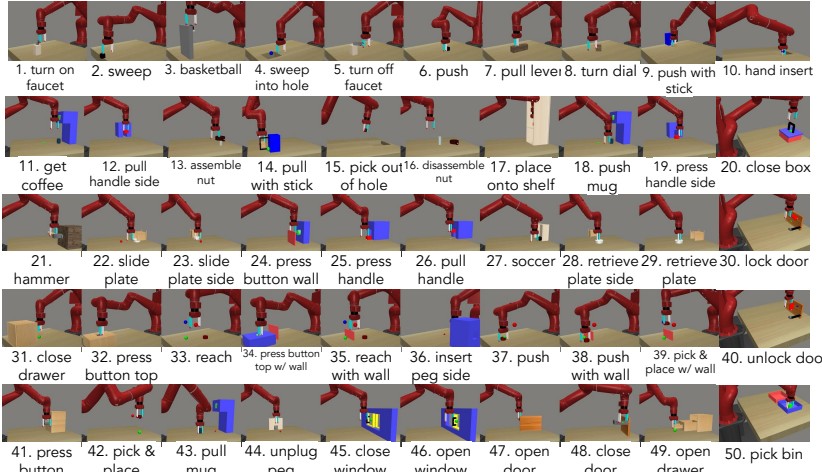

Figure 2: Visualization of the 50 tasks from Meta-World used in the MT50 evaluation. Mint is able to learn about 30 of these tasks.

To answer the above questions, we conduct experiments on multi-task reinforcement learning domains. For multi-task RL doamins, we perform experiments on two multi-task RL benchmark variants MT10 and MT50 (as showcased in Figure 2) proposed by Yu et al. (2019). Finally, to test if Mint can excel in continuous task distributions, we also evaluate the method on a goal-conditioned RL domain where a Sawyer robot arm is required to push a randomly initialized puck to different goal positions. For all RL experiments, we use the popular off-policy RL algorithm, soft actor-critic (SAC) (Haarnoja et al., 2018), which has shown to solve many RL benchmarks with great data-efficiency.

On the multi-task RL domains, we compare Mint to the following methods:

- **SAC**: train a vanilla SAC agent with task identifier as part of the input.
- **Multi-head SAC**: train a SAC agent where both the actor and critic are represented as multi-head feedforward neural networks where the number of heads is the number of tasks.
- **SAC (concat all fc)**: train a SAC agent where the task identifier $z$ is concatenated with the activation at each layer and passed as inputs to the next layer.
- **FiLM (Perez et al., 2018)**: the actor and critic are learned with neural networks combined with FiLM.
- **Superposition (Cheung et al., 2019)**: the actor and critic are learned with neural networks combined with superposition.
- **independent**: learn separate actor and critic per task.

We provide details of architecture design in each domain as well as environment set-up in the Appendix B.

### 6.1 MULTI-TASK REINFORCEMENT LEARNING

On the RL domain, we first investigate the ability of Mint to perform a set of distinct RL tasks. As discussed in Yu et al. (2019), MT10 and MT50 serve as representative benchmarks to evaluate multi-task learning algorithms on learning a diverse range of robotics manipulation tasks. We present the results in Figure 3. The success rates are averaged across tasks and we adopt the success metrics used in the Meta-World benchmark. We design appropriate architectures of the actor and the critic of the SAC algorithm for each method such that the number of parameters of each approach is around the same magnitude (see Appendix B for details).

For MT10, Mint learns all tasks with the best data efficiency, while independent networks also learn all of the tasks with slightly worse data-efficiency. The other methods are unable to acquire half of the skills. Mint, on the other hand, enjoys the expressive power to interpolate between independent learning and sharing, while mitigating optimization challenges, to attain the best results between the two extremes.

For MT50, where the evaluations are done on all 50 of the Meta-World environments, as shown on the right in Figure 3, Mint quickly learns to solve more than $60\%$ of tasks in 20 million environment steps while SAC and SAC with multi-head architectures struggled in solve $40\%$ of the tasks after 35 million steps. Independent networks learn to solve the tasks slower than Mint but eventually surpasses it.[1] This result also validates the expressive power of Mint to represent both separate learning and learning with shared networks.

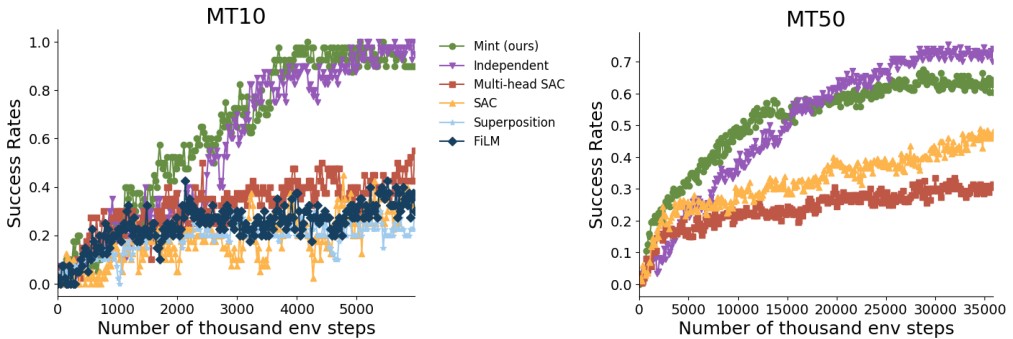

Figure 3: Learning curves on MT10 (left) and MT50 (right). We observe that independent training performs well on both benchmarks. Mint, unlike prior multi-task learning approaches, is able to perform at a similar level to independent training.

We also examine the learned Mint layers in the MT10 setting to analyze whether they capture task-specific information. Specifically, we replace the 'close drawer' task in MT10 with a duplicated 'press button top' task and thus we have two 'press button top' tasks in MT10 (see Figure 2). We

---

[1]We did not evaluate superposition and FiLM on MT50 given the poor performance of these methods on MT10.

train Mint on this new set of tasks. Concretely, if we index the tasks from 1 to 9, then we have two copies of task $T_1$ and learn two separate Mint matrices: $M_{T_1 1}$ and $M_{T_1 2}$.

We then compute the pairwise $\ell_1$ distance between the Mint layer learned for one of the duplicate 'press button top' tasks and the Mint layer learned for any other task such as 'insert peg side' in the set. Specifically, we compute $d(M_{T_1 1}, M_{T_i})$ for all $i \neq 1$, where $d$ is the $l_1$ distance. We compute the percent difference between each $d(M_{T_1 1}, M_{T_i})$ and $d(M_{T_1 1}, M_{T_1 2})$ and present it in Figure 4 on the left. For each pair $M_{T_i}$, $i \neq 1$, we plot $\frac{100 d(M_{T_1 1}, M_{T_i}) - d(M_{T_1 1}, M_{T_1 2})}{d(M_{T_1 1}, M_{T_1 2})}$. From the figure, the two Mint layers learned for the duplicated tasks have the smallest relative $\ell_1$ distance except the distance between the Mint layers of 'press button top' and 'reach', which is reasonable since 'press button top' is structurally similar to 'reaching'.

Finally, we provide a comparison between Mint and other methods where the non-Mint methods are trained with the same number layers as Mint as opposed to the similar number of parameters in Figure 3. As shown on the right in Figure 4, Mint outperforms all of the other approaches in this setting.

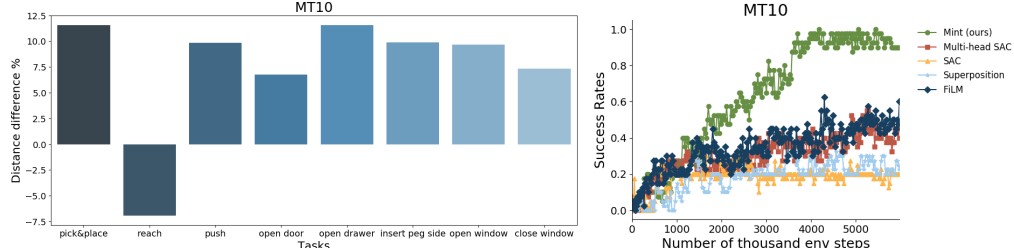

Figure 4: **On the left**, we show the percent increase in $\ell_1$ distance between Mint layers learned for one of the duplicate tasks and each of the other tasks in MT10 as compared to the distance between the Mint layers learned for the two duplicate tasks. We can see that for most of the tasks, the percent increase in $\ell_1$ distance is approximately 10%, except that the distance between 'reach' and 'press button top' is smaller, which could be explained by the fact that 'press button top' is inherently just a reaching task. **On the right**, we compare Mint to a list of other methods with the same number of layers and find that Mint achieves a significantly higher success rate than any of the other approaches.

## 6.2    Goal-Conditioned Reinforcement Learning

Next, we consider the question if Mint can be applied to a set of RL goals that have considerable shared structure. Hence, we evaluate all methods on the goal-conditioned sawyer pushing domain. In this environment, the goal space consists of the initial positions of the puck and the goal positions. For details of the goal-conditioned environment, see Appendix B. At each policy rollout step, we sample a batch of 9 goals and collect 3 paths for each goal, where all the paths are stored in the task-specific replay buffers. At each training step, we sample a batch of 9 goals and 128 samples per goal from their corresponding replay buffers. To prevent creating infinite number of replay buffers, we discretize the goal space into 200 goals. Given that it is impractical to train 200 independent agents, we sample 10 goals from the goal space and train 10 independent SAC agents for estimating the performance of independent training in goal-conditioned pushing. As shown in Figure 5, Mint outperforms all methods both in terms of data efficiency and distance to the goal. SAC (concat all fc) also achieves comparable performance while independent networks fail to learn the task without sufficient amounts of data, suggesting that the ability of Mint to represent both joint training and independent networks per task is crucial in multi-task learning and can lead to considerable improvement.

## 7    Conclusion

Simultaneous optimization of multiple, potentially unrelated tasks can prove challenging for deep neural networks. Recent multi-task learning architectures attempt to mitigate this issue by providing alternative pathways for information to flow through a neural network for each task. In this paper, we introduce a new multi-task learning module, Mint, which provides theoretical guarantees of universal approximation even for multi-task settings with no shared structure. We conjecture that this property, not shared by similar multi-task architectures, enables Mint to outperform other multi-task approaches

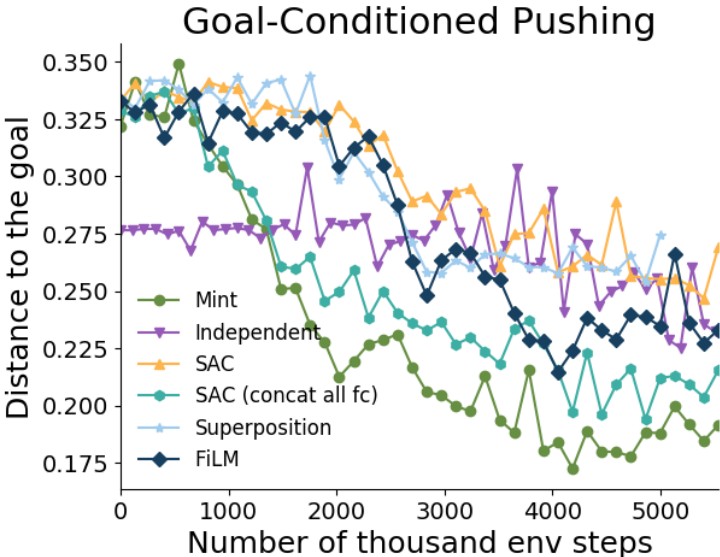

Figure 5: Learning curves on goal-conditioned pushing. Mint is able to outperform all other methods in terms of both distance to the goal and the required number of environment steps.

on a variety of reinforcement learning benchmarks. We also observe that Mint is able to match or improve upon the performance of independent training.

While Mint exhibits strong performance gains over previous methods, one potential limitation is that the task matrices may introduce a significant number of parameters, particularly as the number of tasks increases. As discussed, this can be alleviated for problem domains with many tasks, by learning a single neural network that produces the matrices and biases conditioned on the task descriptor. Further, in our experiments, we find that Mint-based networks can outperform prior methods while using comparable or fewer parameters.

In summary, Mint is a simple, yet effective approach for deep multi-task learning. Its implementation requires minimal modifications over standard deep networks. As a result, we expect it to be straightforward for future work to build upon or use Mint for more effective multi-task learning in deep networks.

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

## A    PROOF OF THEOREM 1

**Lemma 1.** *For a given $\alpha_i$, applying Mint to $y^{(l-1)}$ can express an arbitrary affine transformation at layer $l$ for each task.*

**Proof**

Let $\mathbf{W}^{(l)}$ and $\mathbf{b}^{(l)}$ be an arbitrary weight matrix and bias vector. Suppose that for task $k$ we wish to represent the affine transformation $\mathbf{W_k}^{(l)}y^{(l-1)} + \mathbf{b_k}^{(l)}$ at layer $l$ of the network using the combination of Mint and the affine transformation described by applying $W^{(l)}$ multiplicatively

and $b^{(l)}$ additively. Concretely, we wish to determine whether there exist a task-specific matrix $M(k)^{(l)}$ and task-specific bias vector $\beta(k)^{(l)}$ such that:

$$\mathbf{W_k}^{(l)}y^{(l-1)} + \mathbf{b_k}^{(l)} = W^{(l)}(M(k)^{(l)}y^{(l-1)} + \beta(k)^{(l)}) + b^{(l)} \tag{6}$$

Let $M(k)^{(l)} = W^{(l)^{-1}}\mathbf{W_k}^{(l)}$ and $\beta(k)^{(l)} = W^{(l)^{-1}}(\mathbf{b_k}^{(l)} - b^{(l)})$. Then, the above equality holds.

**Lemma 2.** *(a) For a given $\alpha_i$, there exist affine transformations that cannot be expressed by applying FiLM to $y^{(l-1)}$. (b) Similarly, for a given $\alpha_i$, there exist affine transformations that cannot be expressed by a network which concatenates the input $y^{(l-1)}$ of each layer l with a task indicator $z_k$.*

**Proof of (a).**

We wish show a similar existence proof as in Equation 6, except instead of a per-task weight matrix $M(k)^{(l)}$ at each layer, FiLM uses a per-task modulation vector $\beta(k)^{(l)}$. Concretely, we wish to determine whether there exist a task-specific feature modulation vector $v(k)^{(l)}$ and task-specific bias vector $\beta(k)^{(l)}$ such that:

$$\mathbf{W_k}^{(l)}y^{(l-1)} + \mathbf{b_k}^{(l)} = W^{(l)}(v(k)^{(l)} \odot y^{(l-1)} + \beta(k)^{(l)}) + b^{(l)} \tag{7}$$

where $\odot$ denotes the Hadamard product. This is equivalent to representing $v(k)$ as a diagonal matrix $V(k)$ whose entries along the diagonal are the entries in $v(k)$. Equation 7 is equivalent to

$$\mathbf{W_k}^{(l)}y^{(l-1)} + \mathbf{b_k}^{(l)} = W^{(l)}(V(k)^{(l)}y^{(l-1)} + \beta(k)^{(l)}) + b^{(l)}$$
$$= W^{(l)}V(k)^{(l)}y^{(l-1)} + W^{(l)}\beta(k)^{(l)} + b^{(l)}$$

Equating the corresponding terms on both sides of the equation (those which include $y^{(l-1)}$ and those which do not), equality holds only if:

$$W^{(l)}V(k)^{(l)}y^{(l-1)} = \mathbf{W_k}^{(l)}y^{(l-1)}$$
$$\rightarrow V(k)^{(l)} = W^{(l)^{-1}}\mathbf{W_k}^{(l)}$$

Since this equality must hold for all possible $y^{(l-1)}$

and $\beta(k)^{(l)} = W^{(l)^{-1}}(\mathbf{b_k}^{(l)} - b^{(l)})$. However, since $V(k)^{(l)}$ and $\mathbf{W_k}^{(l)}$ is arbitrary, there is no guarantee that $W^{(l)^{-1}}\mathbf{W_k}^{(l)}$ will be a diagonal matrix.

**Proof of (b).**

Let $\mathbf{W}^{(l)}$ and $\mathbf{b}^{(l)}$ be an arbitrary weight matrix and bias vector. Suppose that for task $k$ we wish to represent the affine transformation $\mathbf{W_k}^{(l)}y^{(l-1)} + \mathbf{b_k}^{(l)}$ at layer $l$ of the network by concatenating a task indicator $z_k$ to $y^{(l-1)}$ and extending $W^{(l)}$ to apply an affine transformation to the concatenated input. This amounts to equating the following transformations:

$$\mathbf{W_k}^{(l)}y^{(l-1)} + \mathbf{b_k}^{(l)} = W_k^{(l)}y^{(l-1)} + b_k^{(l)} + \bar{W}_k^{(l)}z_k$$

where $\bar{W}_k^{(l)}$ is a matrix which is applied to $z_k$. This equation only holds for all possible inputs $y^{(l-1)}$ if $\mathbf{W_k}^{(l)} = W_k^{(l)}$. Since $W_k^{(l)}$ is fixed, we are not able to satisfy this equality.

## B EXPERIMENT DETAILS

We use SAC (Haarnoja et al., 2018) algorithm for all RL domains. For Mint, We use 3-layer fully-connected neural network interleaved with per-task Mint matrices and biases at each layer for both the actor and critic. All the other methods use 6-layer fully-connected neural networks for the actor and critic such that they maintain about the same number of total parameters as Mint. The number of hidden units per layer is 200, 160, and 150 for goal-conditioned pushing, MT10, and MT50 respectively for all methods.

On the goal-conditioned pushing experiment, we sample the $(x, y)$ positions of the puck uniformly from the range $[-0.2, 0.6]$ to $[0.2, 0.7]$ and the positions of the goal uniformly from $[-0.2, 0.85]$ to $[0.2, 0.95]$.

