# OpenReview forum: "Mint: Matrix-Interleaving for Multi-Task Learning"
_ICLR.cc/2020/Conference — Reject_

### Official Review · AnonReviewer3 · 2019-10-23
**Official Blind Review #3**

**Rating:** 3

**Review:**

This paper introduces a factorization method for learning to share effectively in deep multitask learning (DMTL). The approach has some very satisfying properties: it forces all sharable structure to be used by all tasks; it theoretically captures the extremes of total sharing and total task independence; it is easy to implement, so would be a very useful baseline for future methods; and it is able to effectively exploit task similarities in experiments, and outperforms some alternative DMTL methods.

I have two main concerns with the work: (1) It is most closely related to MTL factorization methods, but does not discuss this literature, or provide these experimental comparisons; (2) the interpretation of why Mint works is not clear: it is not clear that the universality is what makes it work, and there are no experimental analyses of what Mint learns.

W.r.t. (1), there are several DMTL approaches that factorize layers across shared and task-specific components, e.g., [1], [2]. Such approaches are extensions of factorization approaches in the linear setting, e.g., [3], [4]. Compared to previous DMTL approaches, Mint is more closely related to these linear methods, as it takes the idea of factorizing each model matrix into two components and applies it to every applicable layer. In particular, the formal definition (i.e., without nonlinear activation between M and W) of Mint appears to be a special case of the more general factorizations in [1]; an experimental  comparison [1] would make the conclusions more convincing, e.g., that universality is important.

However, in the Mint experiments, a non-linear activation is added between the two components of each layer. This could void the universality property. Is there some reason why this is not an issue in practice?

More generally, it is not clear that universality is the important advantage of Mint. Some existing DMTL methods already have this property, including Cross-stitch, which is compared to in the paper. The intriguing difference with Mint is that shared and unshared structure are applied sequentially instead of in parallel. Could there be an advantage in in this difference? E.g., is Mint a stronger regularizer because it forces all tasks to use all shared layers (learning the identity function for shared layers is hard), while something like cross-stitch could more easily degenerate to only use task-specific layers even when tasks are related?

Beyond performance, analysis on what Mint actually learns would be clarifying. Can the sharing behavior be analyzed by looking at the trained Mint layers? Is Mint actually able to learn both of the extreme settings in practice? The non-synthetic experiments in the paper are only performed on tasks that are closely related.

As a final note, adding layers to non-Mint models to make the topologically more similar to Mint models may not help these other models. It may make them more difficult to train or overfit more easily, since they are deeper, but do not have Mint method to assist in training. Comparisons without these extra layers would make the experiments more complete. Do cross-stitch and WPL share the conv layers across all tasks like Mint in Table 1? They should to make it a clear comparison.

Other questions:
-	What exactly are the “two simple neural networks” that produce the goal-specific parameters for goal-conditioned RL? Do these maintain the universality property?
-	Can Mint be readily extended to layer types beyond FC layers? This may be necessary when applying to more complex models.


[1] Yang, Y. & Hospedales, T. M. “Deep Multi-task Representation Learning: A Tensor Factorisation Approach,” ICLR 2017.
[2] Long, M., Cao, Z., Wang, J., & Philip, S. Y. “Learning multiple tasks with multilinear relationship networks”, NIPS 2017.
[3] Argyriou, A., Evgeniou, T., & Pontil, M. “Multi-task feature learning,” NIPS, 2007.
[4] Kang, Z., Grauman, K., & Sha, F. “Learning with Whom to Share in Multi-task Feature Learning,” ICML 2011.


**Experience Assessment:**

I have published in this field for several years.

**Review Assessment: Checking Correctness Of Derivations And Theory:**

I carefully checked the derivations and theory.

**Review Assessment: Checking Correctness Of Experiments:**

I assessed the sensibility of the experiments.

**Review Assessment: Thoroughness In Paper Reading:**

I read the paper at least twice and used my best judgement in assessing the paper.

---

> ### Author Response · Authors · 2019-11-15
> **Author Response to R3**
>
> Thank you for your review! We have uploaded a revised version of the paper to address your feedback and concerns.
>
> 1) Questions regarding tensor factorization-based approaches to multi-task learning.
> Thank you for pointing out this related literature. We have added a discussion and cited all of these methods in Section 5. We also ran experiments with [1] on MT10 and found that it did not perform well, achieving only 10% success rate. We will work to tune the implementation of this method before the final version. We note that, while tensor factorization approaches are general and interesting approaches, Mint is simpler and easier to implement and build upon, which we view as a benefit. Further, Mint is less computationally expensive, as it required about 2-3x less computation.
>
> 2) "The interpretation of why Mint works is not clear: it is not clear that the universality is what makes it work, and there are no experimental analyses of what Mint learns.
> Beyond performance, analysis on what Mint actually learns would be clarifying. Can the sharing behavior be analyzed by looking at the trained Mint layers? Is Mint actually able to learn both of the extreme settings in practice? The non-synthetic experiments in the paper are only performed on tasks that are closely related."
> In light of your feedback, we have performed an additional experiment in a multi-task regime where one task is duplicated. In this case, we show that the task-specific matrices learned by Mint for the two instances of this duplicate task are more similar than the comparison between one of the duplicate task’s matrices and the matrices of other distinct tasks. See Section 6.1 for a more detailed analysis.
>
> 3) "However, in the Mint experiments, a non-linear activation is added between the two components of each layer. This could void the universality property. Is there some reason why this is not an issue in practice?"
> We have corrected the figure in our paper to reflect our implementation of Mint in which the task-specific matrices and shared matrices are not separated by a non-linearity.
>
> 4) "More generally, it is not clear that universality is the important advantage of Mint. Some existing DMTL methods already have this property, including Cross-stitch, which is compared to in the paper. The intriguing difference with Mint is that shared and unshared structure are applied sequentially instead of in parallel. Could there be an advantage in this difference? E.g., is Mint a stronger regularizer because it forces all tasks to use all shared layers (learning the identity function for shared layers is hard), while something like cross-stitch could more easily degenerate to only use task-specific layers even when tasks are related?"
> The usage of a sequential, rather than parallel, flow of data is significant. As the reviewer indicates, it is possible that this architecture might act as a regularizer due to the difficulty of learning to ignore useless shared transformations. In addition, we note the phenomenon observed in residual networks where many residual blocks do not learn useful features and simply converge to the identity function, suggesting that encouraging the network to take advantage of all parallel processing streams is difficult. Using a sequential flow of information prevents this degeneracy.
>
> 5) "As a final note, adding layers to non-Mint models to make the topologically more similar to Mint models may not help these other models. It may make them more difficult to train or overfit more easily, since they are deeper, but do not have Mint method to assist in training. Comparisons without these extra layers would make the experiments more complete."
> We performed such comparisons and added the results to the revised version of the paper (see the plot on the right in Figure 4). Mint still outperforms the baselines without the extra layers in the setting of MT10.
>
> 6) "What exactly are the 'two simple neural networks' that produce the goal-specific parameters for goal-conditioned RL? Do these maintain the universality property?"
> They are 2-layer ReLU networks that take in the goals and return the goal-specific Mint layers. We have added this information to Section 4 in the revised version of the paper. In Lemma 1 in the paper, there are no requirements on the Mint layers, and thus the universality property is still maintained.
>
> 7) "Can Mint be readily extended to layer types beyond FC layers? This may be necessary when applying to more complex models."
> Conceptually, Mint can be readily extended to any type of layer in the sense that we can include blocks of “task specific -> shared” layer for any type of neural network layer. However, the requirement on invertibility (required for universality) is a stronger assumption in layers such as convolutions.

---

### Official Review · AnonReviewer2 · 2019-10-24
**Official Blind Review #2**

**Rating:** 3

**Review:**

The paper propose a single-network approach to multi-task learning by adding a task-specific linear transformation layer after each fully connected layer. The authors prove that the addition of such a layer keeps the expressive power of the network for each task. They also discuss how the linear transformation (parameterized by a transformation matrix and a bias vector) can be represented in a discrete manner in usual multi-task supervised learning and in a continuous manner (by two other neural networks) in goal-conditioned reinforcement learning. Experiments demonstrate the superiority of the proposed single-network approach.

The proposed approach is single and elegant. It is recommended to weak-reject the paper because of the following key reasons.

(1) Problem formulation is far from clear, perhaps because of the lack of clarity in writing. In particular, the super short Section 2 did not clearly illustrate what's being trained and what's being tested, and whether we care about the generalization performance of each task, or the generalization performance to new tasks (generated from P(T)). Some notations are confusing there---for instance, i seems to be indicating tasks, but then there is a z_k as task indicator. Even for the main proposed approach in Section 3.1, the notations are loosely used in nature. For instance, it is hard to understand what the authors mean by "train separate neural networks to output the Mint matrices and biases"---there is no information about the "training data" for learning those neural networks.

(2) Theoretical justification is at best shallow, or at least in the context that the authors have put it. While having an universal expressive power is good, it is easily achieved by adding an indicator variable (z_k) per layer (similar to task-specific-all-fc in the experiments). So the guarantee does not seem to be closely related to explaining the proposed approach (though the guarantee is nice to have). The authors contrast the guarantee with what FiLM (a competitor approach) can do, but in the experiments FiLM is not taken as a competitor in multi-task supervised learning, leaving a big gap between theory and practice. In the flow presented by the authors, it is strongly suggested to introduce FiLM in more detail and compare it with the proposed approach more clearly in design, theory and experiments.

(3) It is hard to understand whether the experiments are reasonably designed. In particular, the two settings take different sets of competitors, and there is little information on why those competitors are selected, whether they represent state-of-the-art, etc.. The authors highlight that the proposed approach uses much fewer parameters but other than that it is hard to infer why the proposed approach is better. Is it better because there is more overfitting for the competitor's approaches given more parameters? Is it better because it is easier to tune? The task-specific-all-fc (which is of similar # parameters to the proposed approach) result particularly looks suspicious to me but there is no other information to double-check on why the proposed approach is better. In particular, I believe the authors have *not* answered their first proposed question "does our method enable effective multi-task learning both in settings where there is substantial overlap in tasks and where there is little overlap" properly---their best evidence may have been MT10 and MT50 experiments, but even in those experiments, I am not sure whether the authors want to take the results as suggesting there are "substantial overlap" or "little overlap."

Some other suggestions:

(4) It is suggested to analyze the matrices learned by the proposed approach. Do the matrices contain reasonable task correlations (i.e. for two similar tasks, are the matrices somewhat similar) to understand more about the proposed approach.

(5) It looks a bit strange to me that there is no discussion on regularizing the linear transformation matrices, as it seems possible to embed the task relations through the regularization. Have the authors considered the possibility?

(6) The authors are overly-emphasizing what they want to do (interpolating between independent networks and shared network). This occupies multiple redundant paragraphs in the early sections. It is better to remove some of those and use the space for more solid results, such as clarifying the notations.

(7) One baseline that could have been considered is to just train a fully-shared network (without z_k), and a fully-independent one. Then, use validation to select the better network and compare with the proposed approach.


**Experience Assessment:**

I do not know much about this area.

**Review Assessment: Checking Correctness Of Derivations And Theory:**

I assessed the sensibility of the derivations and theory.

**Review Assessment: Checking Correctness Of Experiments:**

I assessed the sensibility of the experiments.

**Review Assessment: Thoroughness In Paper Reading:**

I read the paper at least twice and used my best judgement in assessing the paper.

---

> ### Author Response · Authors · 2019-11-15
> **Author Response to R2**
>
> Thank you for your review. We have uploaded a revised version of the paper to address all of your concerns. Below, we address the feedback that you provided.
>
> (1) “Section 2 did not clearly illustrate what's being trained and what's being tested, and whether we care about the generalization performance of each task, or the generalization performance to new tasks (generated from P(T)).”
> Thank you for pointing out this lack of clarity. We have revised Section 2 to explain that we care about performance across all the tasks that we train on. Specifically, we have a fixed set of K tasks, and we wish to obtain high performance across all of these K tasks. We do not care about generalization performance.
>
> “Some notations are confusing there--for instance, i seems to be indicating tasks, but then there is a z_k as task indicator.”
> We have changed the notations so that the tasks are denoted by T_k and the task indicators are z_k. The distinction between T_k and z_k is that T_k is the task itself whereas z_k is an indicator of the task that is provided as input to a neural network.
>
> (2) “While having an universal expressive power is good, it is easily achieved by adding an indicator variable (z_k) per layer (similar to task-specific-all-fc in the experiments). So the guarantee does not seem to be closely related to explaining the proposed approach.”
> Adding an indicator variable per layer is not sufficient for achieving universal expressive power. Specifically, using task-specific weights in this way amounts to adding task-specific bias terms to the activations of the network which processes the inputs. We have extended the theory (Lemma 1 and its proof) to the case where a task indicator is added to each layer to explain why this does not achieve universal expressive power.
>
> “It is strongly suggested to introduce FiLM in more detail and compare it with the proposed approach more clearly in design, theory, and experiments.”
> We have added a more in-depth discussion and theoretical comparison of Mint, FiLM, and task indicator conditioning in Section 3.2, and we have a direct empirical comparison to FiLM in the MT10 experiments (see Figure 3).
>
> (3) "I believe the authors have *not* answered their first proposed question "does our method enable effective multi-task learning both in settings where there is substantial overlap in tasks and where there is little overlap" properly. "
> We agree that defining “little overlap” and “substantial overlap” between tasks is difficult. In our RL experiments, we selected MT10 and MT50 to highlight multi-task learning with relatively little overlap, as the agent must learn distinct skills. In contrast, we selected the goal-conditioned pushing environment as a multi-task learning environment with relatively larger overlap between tasks. In both experiments, we observed the benefits of Mint over other multi-task learning approaches.
>
> (4) "It is suggested to analyze the matrices learned by the proposed approach. Do the matrices contain reasonable task correlations?"
> To perform this analysis, we ran a multi-task experiment among a set of tasks where two tasks are exactly the same, and compared the matrices learned for these two tasks, in comparison to those from two different tasks. Specifically, we computed the L1 norm of the difference between the two matrices of the same task and compared that with the L1 norm of the difference between two task matrices corresponding to different tasks. We have added this analysis to Section 6.1 (see Figure 4).
>
> (5) "It looks a bit strange to me that there is no discussion on regularizing the linear transformation matrices...Have the authors considered the possibility?"
> We experimented with regularizing the linear transformation matrices of Mint by maximizing the pairwise cosine distance between them. We found that this regularization did not impact the performance of Mint.
>
> (6) "The authors are overly-emphasizing what they want to do (interpolating between independent networks and shared network). This occupies multiple redundant paragraphs in the early sections. "
> We reduced the redundancy in Section 3.1.
>
> (7) "One baseline that could have been considered is to just train a fully-shared network (without z_k), and a fully-independent one."
> In the MT10, MT50, and goal-conditioned pushing experiments, we used a fully-shared network (SAC) and a fully independent one (independent) and compared these methods to Mint. See Figures 3 and 4.

---

### Official Review · AnonReviewer1 · 2019-10-29
**Official Blind Review #1**

**Rating:** 3

**Review:**

This paper proposes a matrix-interleaving (Mint) based on neural networks for multi-task learning. The Mint contains a share parameter matrix and a task-specific parameter matrix.

Authors claim that the proposed Mint have the ability to represent both extremes: joint training and independent training. However, if the relations among tasks make the model between the both extreme cases, how is the performance of the proposed model?

What does "when the shared weight matrices are not learned" mean? Are the shared weight matrices randomly initialized and then fixed without updating?

Theorem 1 requires that each W^(l) is invertible, which implies that W^(l) is a square matrix. This requirement may not be satisfied in many neural networks. In this case, does Theorem 1 still hold? If not, Theorem 1 is not so useful.

**Experience Assessment:**

I have published in this field for several years.

**Review Assessment: Checking Correctness Of Derivations And Theory:**

I assessed the sensibility of the derivations and theory.

**Review Assessment: Checking Correctness Of Experiments:**

I assessed the sensibility of the experiments.

**Review Assessment: Thoroughness In Paper Reading:**

I read the paper at least twice and used my best judgement in assessing the paper.

---

> ### Author Response · Authors · 2019-11-15
> **Author Response to R1**
>
> Thank you for your review. Below, we address the feedback that you provided.
>
> “If the relations among tasks make the model between the both extreme cases, how is the performance of the proposed model?”
> In our goal-conditioned pushing experiments, the relation among the tasks is between the extreme cases. In this setup, Mint outperforms independent training and performs slightly better than joint training.
>
> “What does ‘when the shared weight matrices are not learned’ mean?”
> We have removed this statement to avoid confusion and revised the text. What we meant by this statement was that even if the shared weight matrices are no longer changing (e.g. they have been fully optimized), there exist task-specific Mint layers which can allow the Mint network to express the same transformations of the input as an optimal task-specific network.
>
> “Theorem 1 requires that each W^(l) is invertible, which implies that W^(l) is a square matrix. This requirement may not be satisfied in many neural networks. In this case, does Theorem 1 still hold? If not, Theorem 1 is not so useful.”
> In practice, we can design the Mint network such that W^(l) is always an MxM square matrix for some M (M can be different for each layer), and thus satisfy the conditions of the theorem. Specifically, we can first apply a Mint layer which consists of an MxN weight matrix and then apply the shared fully-connected layer which consists of an MxM weight matrix. Therefore, the Mint layer will transform an N-dimensional input to an M-dimensional output, and the shared network contains only square matrices.

---

### Official Review · AnonReviewer4 · 2019-11-01
**Official Blind Review #4**

**Rating:** 6

**Review:**

In this paper, the authors propose a simple but effective matrix-interleaving method (mint) for multi-task learning, which aims to represent both joint training and independent training.
The model achieves good performance on several supervised and reinforced learning datasets. Though the model resembles FiLM(Perez et al., 2018), it outperforms FiLM by a larger margin in three dataset.

It would be better for authors to give more detailed comparisons with models that work on combining both joint training and independent training.

I would like to see the paper to be accepted for its simplicity and effectiveness.

Typos: chnages -> changes?

**Experience Assessment:**

I do not know much about this area.

**Review Assessment: Checking Correctness Of Derivations And Theory:**

I assessed the sensibility of the derivations and theory.

**Review Assessment: Checking Correctness Of Experiments:**

I assessed the sensibility of the experiments.

**Review Assessment: Thoroughness In Paper Reading:**

I read the paper at least twice and used my best judgement in assessing the paper.

---

> ### Author Response · Authors · 2019-11-15
> **Author Response to R4**
>
> Thank you for your comments. We have uploaded a revised version of the paper that addresses your concerns.
>
> Regarding methods that use both joint and independent training, we note that for our RL experiments, we compare with superposition, which is a method that combines joint training and independent training and find that Mint outperforms superposition in both MT10 and goal-conditioned RL (see Figure 3 and Figure 5). We have also made various stylistic and typo fixes to improve readability of the text.

---

### Author Response · Authors · 2019-11-15
**Response to all Reviewers**

To address the reviewers’ concerns, we have ran several new experiments & made several updates to the paper listed below:
(R1, R2) Improvement in clarity and removal of redundancy
(R3) Comparison to shallower network
(R3) Discussion of tensor factorization approaches
(R2) More detailed discussion of and comparison to FiLM and task-indicator conditioning
(R2, R3) New analysis of learned task matrices

Lastly, we discovered a bug in the CIFAR experiments, derived from the open-source implementation of routing networks. The bug was that methods were being trained for 3 epochs instead of 50. We unfortunately did not have time to rerun and verify all of the CIFAR results in time for the paper revision, so we are omitting those experiments from the current revision of the paper. We will add these experiments to the final version of the paper, once completed.
Even without the results on CIFAR, we believe the goal-conditioned RL experiment, the two multi-task RL experiments, as well as the new analysis and additional comparisons, sufficiently illustrate the merit of Mint.

---

### Decision · Program_Chairs · 2019-12-19

**Decision:**

Reject

**Comment:**

Reviewers put this paper in the lower half and question the theoretical motivation and the experimental design. On the other hand, this seems like an alternative general framework for solving large-scale multi-task learning problems. In the future, I would encourage the authors to evaluate on multi-task benchmarks such as SuperGLUE, decaNLP and C4. Note: It seems there's more similarities with Ruder et al. (2019) [0] than the paper suggests.

[0] https://arxiv.org/abs/1705.08142